# Dendrimers in Alzheimer’s Disease: Recent Approaches in Multi-Targeting Strategies

**DOI:** 10.3390/pharmaceutics15030898

**Published:** 2023-03-10

**Authors:** Cécile Arbez-Gindre, Barry R. Steele, Maria Micha-Screttas

**Affiliations:** Institute of Chemical Biology, National Hellenic Research Foundation, 11635 Athens, Greece

**Keywords:** Alzheimer’s disease, dendrimers, drug delivery, blood–brain barrier, Aβ amyloid, tau protein, neuroinflammation, mitochondrial dysfunction, oxidative stress

## Abstract

Nanomaterials play an increasingly important role in current medicinal practice. As one of the most significant causes of human mortality, and one that is increasing year by year, Alzheimer’s disease (AD) has been the subject of a very great body of research and is an area in which nanomedicinal approaches show great promise. Dendrimers are a class of multivalent nanomaterials which can accommodate a wide range of modifications that enable them to be used as drug delivery systems. By means of suitable design, they can incorporate multiple functionalities to enable transport across the blood–brain barrier and subsequently target the diseased areas of the brain. In addition, a number of dendrimers by themselves often display therapeutic potential for AD. In this review, the various hypotheses relating to the development of AD and the proposed therapeutic interventions involving dendrimer–base systems are outlined. Special attention is focused on more recent results and on the importance of aspects such as oxidative stress, neuroinflammation and mitochondrial dysfunction in approaches to the design of new treatments.

## 1. Introduction

Alzheimer’s disease (AD) is a complex, progressive, and irreversible neurodegenerative brain disease characterized by a variety of symptoms such as loss of memory, decline of cognitive functions, abnormal behavior and psychiatric problems [1,2]. It is the most frequent age-related cause of dementia in Western society and has become a public health problem with great social and economic consequences, particularly for the families and caregivers of patients, since it affects and will continue to affect a large part of the population over the age of 65. It is currently estimated that dementia affects 55 million people worldwide and that this number will increase by 10 million new cases each year. AD is considered to contribute to dementia in 60–70% of these cases [3]. 

Symptomatic mitigation of AD using strategic interventions can temporarily improve cognition and memory, but no effective therapy has yet been found which is able to arrest or delay the progression of this disease. The majority of research efforts are directed towards the development of a combination of therapeutic approaches and, since a preclinical stage of AD generally precedes the disease by a few decades, the focus of these approaches is on treatments that can inhibit or delay the progression of the disease before it becomes irreversible [4]. 

The complexity of this disease is highlighted by the wide range of new drugs that have been developed for the potential treatment of AD [5,6], but which have failed to demonstrate significant efficacy in phase III clinical trials. It appears that a full understanding of the real physiopathological and biological mechanisms of AD is still a long way off [7,8]. Nevertheless, one crucial aspect which has been underlined by a plethora of research studies concerns access to the central nervous system (CNS) and the brain via the safe and effective transport of drugs across the blood–brain barrier (BBB) [9]. The BBB prevents the entry of harmful substances into the brain, and the design of suitable means for efficient drug delivery plays a major role in the development of therapeutic approaches towards AD as well as other neurodegenerative conditions. Many of the proposed strategies involve the use of nanomaterials, and a great body of knowledge has now been accumulated concerning the development of nanocarriers that are able to improve the stability and efficacy of hydrophobic drugs and have the capacity to cross the BBB without damage [10]. Subsequent penetration of the brain and the targeted distribution of therapeutics can thereby potentially improve local accumulation efficacy and reduce adverse effects due to the high dosages that would otherwise be needed. There is consequently much current research into nanotechnological platforms for the treatment of AD, which complement conventional approaches and provide possible solutions to existing challenges.

In this review, after an overview of current hypotheses concerning the etiology of AD and a brief outline of the structure of the BBB, we focus on some of the strategic nanotechnological therapeutic approaches to AD involving dendrimers which have been developed to facilitate drug delivery across the BBB or which show promise as drugs per se. In view of the fact that carrier or receptor binding structures for crossing the BBB will be the main transport target for drug delivery systems, special consideration is given to the importance of the appropriate design of the dendrimer interface and how this design can be facilitated by the various features that dendrimers offer. Approaches from the most recent research studies will be given particular attention.

## 2. Recent Advances in Hypotheses for the Etiology of Alzheimer’s Disease

AD is predominantly associated with the aging population, but 10% of cases are diagnosed before the age of 65 (so-called early-onset AD, EOAD), and these have a genetic etiology of up to 100%. Late-onset AD (LOAD or sporadic AD) affects the remaining 90% of patients. Apart from aging, other risk factors include sex, diabetes, hypertension, obesity, hypercholesterolemia and traumatic brain injury [1,2]. 

Although Alzheimer first clinically described AD in 1906, the molecular identities of its two defining pathologies, namely the deposition of amyloid-β-peptide (Aβ) in the form of β-sheet rich fibril plaques in the extracellular space and the hyperphosphorylated tau protein found in the intraneuronal neurofibrillary tangles (NFTs), were not characterized until the mid-1980s. Particular regions of the brain in AD patients display an abundance of these two abnormal protein structures and loss of connections between cells. While this is also a phenomenon of normal aging, these two factors are found in abnormally large quantities in patients suffering from AD. 

It is now recognized that the symptomatic phase of AD is preceded by a long period during which the increase in senile plaques and NFTs is not yet significant. This latency is characterized by an intermediate phase of mild cognitive impairment (MCI) which can last up to a few decades and is accompanied by a significant oxidative imbalance. It has been shown that individuals with MCI have a higher risk of progressing to early-onset AD. AD is associated with dysfunction of the cholinergic and dopaminergic systems. In the former, reduced levels of acetylcholine (ACh) are observed together with a decrease in the activity of acetylcholinesterase (AChE) and an increase in the activity of butyrylcholinesterase (BuChE). As a result, cholinesterase inhibitors have received much attention as possible drugs. In the dopaminergic system, different degrees of dopamine dysfunction can also occur in the course of all phases of AD, and have been linked to the presence of oxidative stress. Metal dyshomeostasis (copper, iron and zinc), metal-induced oxidative stress, neural Ca^2+^ dysfunction or calcium dyshomeostasis, and pathologies such as neuroinflammation and mitochondrial dysfunction related to protein aggregation have also been strongly connected with AD. All of the above pathological manifestations have given rise to a number of hypotheses, which provide an important starting point for the design and discovery of potential therapeutic approaches for AD, and are outlined in the following sections [7].

### 2.1. Amyloid Hypothesis 

The neuropathology of AD is manifested macroscopically by brain shrinkage with cortical thinning and atrophy, while the main microscopic features include the formation of amyloid plaques consisting of large accumulations of Aβ and NFTs. Aβ is a sticky brain protein fragment that accumulates and aggregates in the CNS. It is implicated in normal aging, but numerous human disorders, such as AD, Parkinson ’s disease, type II diabetes and Creutzfeldt–Jacob’s disease, are also usually associated with the pathological aggregation of peptides or proteins, [11,12]. 

While plaques are characteristic of AD, the plaque burden is not correlated with cognitive impairment, but rather with the neurocytotoxicity of soluble low-molecular-weight (LMW) Aβ oligomers which are formed upstream of plaques. Aβ oligomerization seems to be the first change that occurs biochemically in the development of AD [8]. Aβ is a product of the processing of the transmembrane amyloid-β precursor protein (APP) and, according to the amyloid hypothesis [13], its involvement is postulated to be a major factor in the development of AD. By itself, APP is not neurotoxic and does not produce Aβ until it undergoes abnormal sequential proteolytic cleavage of its Aβ domain, which can occur via either a non-amyloidogenic or an amyloidogenic pathway (Figure 1). 

AD development is only observed as a result of the latter pathway in which sequential cleavage by β- and γ-secretase yields several isoforms of the Aβ protein monomer. Mutations in genes which encode for APP and presenilins, catalytic subunits of γ-secretase, are implicated in certain types of early-onset AD [15]. The cleavage of APP predominantly yields peptides containing 40 amino acids, and Aβ40 is found to have a particular tendency to deposit on the walls of the vessels around the cerebral vasculature, leading to cerebral amyloid angiopathy. Longer Aβ42 fragments are also produced and have a greater propensity to self-aggregate into soluble toxic oligomers that ultimately coalesce faster into extracellular plaques or fibrils (Figure 2). Aβ42 oligomers are more effective seeds for fibril growth than monomers or mature fibrils [16]. Due to their progressive accumulation and overproduction, or a failure in the clearance mechanism, these normally soluble monomers can self-assemble to give rise to a whole range of LMW toxic oligomers that rearrange into high-molecular-weight non-fibrillar species or protofibrils as well as mature insoluble fibrils. Although in vitro experiments cannot accurately represent what is happening in the brain, it is generally agreed that soluble oligomers are the primary cytotoxic species and are much more cytotoxic than protofibrils and fibrils. Neuronal damage begins to occur several decades before the onset of Alzheimer’s disease symptoms, and some studies have shown that, by binding to receptors on the neuronal membrane, oligomers can damage its integrity. They can then enter the cell and initiate cell damage and cell death through mitochondrial damage, endoplasmic reticulum stress, calcium ion dysregulation and apoptosis [8,17]. 

In view of the proposed involvement of Aβ peptides in the main self-assembly processes, the inhibition of fibrillation represents one of the approaches for potential therapeutic interventions. Structurally, the Aβ peptides display a hydrophobic cross-linked beta sheet morphology containing hairpin fragments aligned in parallel and stabilized by intermolecular interactions, while the connection of the β strands via a hydrophilic region imposes the curvature found in both oligomers and fibrils [12]. As a result of these findings, strategies using beta-turn peptidomimetic conjugates as inhibitors of the aggregation of amyloid peptides have recently been developed [18].

The amyloid hypothesis has influenced and directed drug development for many years. Secretase inhibitors have been developed and tested with the aim of suppressing the formation of Aβ, but the results have not been fully convincing [19,20]. Promising results have been obtained with anti-Aβ strategies using human monoclonal antibodies such as aducanumab and lecanemab, which reduce Aβ and slow down cognitive decline, although serious side-effects can occur [21,22]. Despite the continued search for effective anti-Aβ therapeutics, however, it is still unclear whether the removal of amyloid by itself can provide a cure for AD, and there has more recently been a shift toward exploring alternative approaches.

A variation on the amyloid hypothesis has been proposed which implicates the involvement of Aβ in the formation of ion channels by soluble Aβ oligomers [23]. There is substantial experimental evidence that Aβ peptides, and Aβ42 in particular, possess structural features that promote the formation of ion channels in cell membranes which lead to the disruption of calcium ion homeostasis and subsequent neuron cell apoptosis [24,25]. Based on these observations, it has been suggested that these peptides may be responsible for some or all of the observed neurotoxicity. Support for the ion channel hypothesis comes from the observation that drugs designed to target the Aβ fibrillization and plaque formation process have not shown notable effectiveness. It has also been observed that metals such as cadmium, mercury and lead are possible risk factors for AD, and it is postulated that these metals interact with the Aβ42 ion channel and affect its biophysical parameters [26,27]. Therapies based on the ion channel hypothesis may therefore reveal new targets for the development of alternative therapeutic approaches to AD.

### 2.2. Tau Hypothesis

AD belongs to a class of neurodegenerative diseases called tauopathies, which are characterized by the deposition of abnormal tau protein in the brain [28,29]. Tau is a multifunctional protein involved in the stabilization of microtubules and, because it plays a critical role in AD, attention has been focused both on the protein itself and its phosphorylation. In AD, tau becomes hyperphosphorylated and is unable to bind the microtubules, and it begins to aggregate forming intracellular NFTs in neurons and astrocytes (Figure 3). In contrast to the lack of correspondence between plaque burden and dementia, there is a stronger case for a link between NFTs and disease progression. Indeed, the amount and distribution of NFTs have been shown to correlate with both the severity and duration of dementia [30]. 

### 2.3. Neuroinflammation Hypothesis

Researchers generally agree that neuroinflammation and AD pathogenesis are closely associated, with one being the cause or the effect of the other, independently of what is triggering what [31]. In diseases such as AD, misfolded or aggregated proteins (Aβ, NFT) trigger neuroinflammation through the activation of microglia which mediate the innate immune capacity of the CNS and perform primary immune surveillance, as well as through the expression of proteic complexes such as inflammasomes which are crucial for innate immunity [32]. Indeed, the inflammasome NLRP3, whose connection with neuroinflammation has been widely characterized, has been particularly identified as a potential therapeutic target for the treatment of AD [33,34] (Figure 4). 

The activation of microglia can result in either beneficial or harmful effects throughout the beginning and progression of AD. The early activation of microglia in AD can offer neuroprotection by promoting the clearance of Aβ, whereas by means of an alternative pathway proinflammatory cytokines are produced, resulting in the down-regulation of Aβ-clearing agents and the corresponding accumulation of Aβ aggregates. A number of drugs have been examined in connection with the neuroinflammatory hypothesis but, so far, none has been shown to give significant beneficial results.

The normal effect of any infection is to cause inflammation and oxidative stress and, in connection with AD, much attention has been directed towards pathogens such as human herpes simplex virus-1 (HSV-1) since the presence of HSV-1 antibodies in the brain was positively correlated with AD pathology [31,35,36,37,38]. Although it is still debated whether viruses such as HSV-1 play a causative role in the development of AD, clinical studies have shown that treatment with antiherpetics or other antivirals can lead to a reduction in the incidence of dementia as well as to an improvement in cognitive function in AD patients. The actual process by which HSV-1 is involved in the development of AD is still open to debate, but there is evidence for its involvement in all three of the hypotheses for the etiology of AD. In addition to the inducement of AD by the inflammatory response due to HSV-1, it has been postulated that overproduction of Aβ due to the repeated cycle of reactivation for HSV-1 could lead to the accumulation of Aβ in the brain and activate the pathogenic amyloid pathway. Alternatively, infection with HSV-1 is known to increase the hyperphosphorylation of tau protein, and this could provide another pathway for AD pathology [36].

### 2.4. Oxidative Stress 

Under normal conditions, levels of reactive oxygen species (ROS) in the body are regulated by endogenous antioxidant defenses such as detoxifying enzymes (superoxide dismutase, catalase and glutathione peroxidase). When this defensive capacity is reduced, however, an imbalance in ROS levels gives rise to oxidative stress, causing adverse effects on the redox chain such as a decrease in ATP production. Energy and oxygen consumption is high in the brain, while it is also rich in catalytic iron ROS. The lack of antioxidant-related enzymes, however, together with readily oxidizable lipids, makes the CNS very vulnerable to ROS. Thus, while oxidative stress may be a typical process in aging, it is considered to be integral to a range of age-dependent neurological disorders including AD [7,39]. It has been hypothesized that metal-induced oxidative stress plays an important role in the pathology of AD, and evidence to support this has been obtained from the precise quantification of redox-active transition metals in brain tissues, where levels of copper, zinc and iron are found to be many times the levels found in normal tissues [40]. 

### 2.5. The Mitochondrial Cascade Hypotheses and Mitochondrial Dysfunction

Major neurodegenerative diseases are characterized by the accumulation of misfolded proteins or by genetic mutations which have been associated with mitochondrial dysfunction [41,42]. Mitochondria are major eukaryotic cytoplasmic organelles with many functions. They are mainly involved in energy production through the oxidative phosphorylation pathway, in the production of metabolic substrates, as well as in cell proliferation or programmed cell death, calcium homeostasis, and inflammation [43]. Mitochondria provide ATP through the electron transport chain and are responsible for the generation of 90% of endogenous ROS [44,45,46,47]. Under physiological conditions, mitochondrial quality control (MQC) processes are able to repair damage due to ROS but, if these processes break down, the rate of development of symptoms of AD can accelerate. Indeed, abundant structurally damaged mitochondria have been found in the brains of biopsied AD patients.

In contrast to the role of ROS, inflammatory responses associated with mitochondrial dysfunction have been less widely studied. Mitochondria have many similarities to bacteria and, in the CNS, the escape of mitochondrial content initiates pro-inflammatory immune responses in glial cells, thereby leading to chronic neuroinflammation and the progression of the pathology of neurodegenerative diseases. 

The close link between mitochondrial function and inflammatory signals and AD symptoms and pathogenesis has led to the proposal of the primary and secondary mitochondrial cascade hypotheses. According to the primary mitochondrial cascade hypothesis, mitochondrial dysfunction exists independently and upstream of Aβ deposition and is the trigger for plaque formation in the brains of AD patients. Conversely, certain molecular mechanisms seem to support a secondary mitochondrial hypothesis in which ROS-induced mitochondrial dysfunction is influenced by upstream pathologies including Aβ [41,47].

## 3. The Blood–Brain Barrier: Structure and Transport Mechanisms 

The CNS is made up of a series of barriers, among which the blood–brain barrier (BBB) plays a specific role in the protection of the brain from neurotoxic molecules. By maintaining CNS homeostasis through its involvement in the regulation of the entrance and discharge of ions, cells, and small and large molecules or macromolecules between blood and brain tissue, the BBB is involved in the careful maintenance of a microenvironment for neuronal signaling [48,49]. Alterations in the structure of the BBB are seen in chronic neurodegenerative diseases, and it is still uncertain as to whether this is a downstream process or whether it plays a significant role in disease onset and development [50].

The BBB mainly comprises microvascular endothelial cells (BMECs) forming the walls of blood vessels through intermolecular tight junctions. BMECs monitor the motion of molecules and ions from their luminal plasma membranes (facing the blood compartments) to the abluminal membranes directed toward the extracellular fluid (facing the brain) (Figure 5).

Since BMECs are lipophilic, substances in the hydrophilic environment of the blood have difficulty in passing through. Direct transcellular crossing through the endothelial membrane is allowed only for small lipophilic compounds with a molecular weight under 400–500 Da. Therefore, the transport of molecules occurs principally via transcellular transporters, while efflux pumps situated on the membrane of BMECs act as agents for the clearance and return into circulation of toxic and other unwanted substances. Nutrients and essential compounds cross the BBB by making use of carrier-mediated transporters (CMT). Their binding to these specific transporters on the surface of the membrane triggers conformation changes, which allow passage through the barrier. For instance, GLUT1, a glucose-transporter, enables glucose uptake, and it is worth noting here that a recent study associated neurodegeneration with reduced levels of GLUT1 in the brain. Luminal and abluminal membranes of BMECs also present various other agents involved in the transport of amino acids. Amino acid transporters such as system-L are involved in the transport of valine and histidine, while ASC transporters are involved in the transport of alanine, serine or cysteine. Another means of penetration of the BBB is via absorptive-mediated transcytosis (AMT). This has low specificity but is characterized by a high transport capacity from the luminal to the abluminal surface. This mechanism of internalization is triggered by electrostatic interactions of positively charged compounds such as proteins with the anionic membrane of the BBB, which has a high affinity toward cationic molecules. Finally, the receptor-mediated transcytosis (RMT) mechanism is used by large endogenous molecules such as neuropeptides, proteins, and hormones, which form specific complexes that trigger an exocytosis process for crossing the endothelial “wall”. These mechanisms are illustrated in Figure 6.

The BBB provides a physical, physiological and biochemical barrier that segregates the circulating blood compartment from the cerebral parenchyma, and almost 100% of large-molecule neurotherapeutics and >98% of small-molecule drugs are denied direct access to the brain. This poses difficulties for normal drug delivery methods such as the oral dosing of specific drugs, neuropharmaceutical agents, genes, proteins or peptides. The manner by which a molecule crosses the BBB is a function of its physicochemical properties, namely size, lipophilicity and charge, and the regulatory aspects of the BBB have a significant impact on the pharmacotherapy of AD. One of the challenges and priorities of pharmaceutical research therefore relates to the efficient delivery of drugs, namely the controlled and localized delivery to a target site in order to reduce drug toxicity and to increase effectiveness [9,49]. 

Drugs rarely fulfill the lipophilicity, charge or size requirements for passive transport across the BBB. Many therapeutics have hydrophilic properties, and approximatively 98% of small active compounds and all large molecules (MW > 400 Da), such as recombinant proteins, peptides or gene-based medicinal, are blocked by the BBB. This has led to the development of specific invasive and non-invasive approaches to bypass the BBB and to ensure safe and effective drug delivery to the brain. Invasive methods (e.g., ultrasound, radiotherapy, biochemical or osmotic disruption) disrupt the BBB’s tight junctions and enable it to be bypassed. Since this can entail a high risk of damage to the integrity of the brain, they have only been applied in limited situations such as for brain tumors or traumatic injuries. Invasive surgery also permits the local delivery of drugs, but such approaches involving the modification of BBB permeability can give rise to complications such as neuronal dysfunction or inflammation induced by the leakage of membrane proteins or the entry of toxins or pathogens [50]. In order to circumvent the problems associated with these invasive methods, numerous non-invasive methods have been examined. Among them, modifications of conventional drugs have stimulated the recent development of nanotechnology platforms for crossing the BBB to enable drug delivery to the CNS [10]. With the goal of improving blood to brain transport, nanoplatforms have incorporated modifications such as cationization in order to trigger the AMT process, modifications with small nutrients such as amino acids, hexoses, or vitamins to activate CMT, or the incorporation of endogenous large molecules such as transferrin, lactoferrin, neuropeptides, lectins, insulin, or insulin-like growth factor to activate RMT. Although the pore size in the BBB allowing a passive diffusion is usually less than one nanometer, nanoparticles that have a diameter of several nanometers can cross the BBB by means of CMT [53]. 

Various types of nanoparticles (NPs) (liposomes, polymeric nanoparticles, solid NPs, micelles, dendrimers) have been examined as drug delivery systems (DDS) for enhancing the efficacy of agents administered to the brain. As a result, there is a now a rich body of literature available concerning recent developments in promising nanotechnological approaches concerning AD [54,55]. Despite the recent European rejection of the drug aducanumab, due to its harmful side effects, the first clinical success of this monoclonal antibody, which removes amyloid plaques in the brain and slows the progression of the disease, paved the way for the development of strategies using dendrimers both as DDS and as drugs per se [53].

Dendrimers are NPs of small size which represent promising candidates for non-invasive approaches and for systemic drug administration and delivery to the brain. Their special characteristics not only enable them to be modified in order to be transported through biological barriers, including the BBB, but also allow them to play a dual role as therapeutic agents and nanocarriers. As a result, not only are dendrimers potentially versatile nanocarriers with therapeutic and diagnostic applications, but also specific types of dendrimers have been shown by themselves to exert intrinsic anti-amyloidogenic effects [53]. But what precisely are dendrimers?

## 4. The Nature of Dendrimers and Their Biomedical Applications

Dendrimers are a special kind of hyperbranched polymer, the term having been coined by Tomalia from the Greek “*dendron*” and “*meros*”, meaning “tree” and “part”, respectively, to describe a globular nanostructure reminiscent of a tree with a size ranging between 10 and 100 nm [56,57]. Dendrimers display a distinctive molecular architecture consisting of a central core from which radiate repeating branching units, the number of which defines a dendrimer’s “generation” (denoted as G0, G1, G2, G3, etc.), and whose termini constitute the periphery or surface of the dendrimer (Figure 7). The interior of the dendrimer contains cavities which become more significant in the higher generation materials.

All parts of the dendrimer (core, branching units and terminal groups) influence its physicochemical properties, such as the size, hydrophilicity, surface charge, functionality and conformational flexibility. These properties can be tuned through the appropriate chemical manipulation of these parts. As a result, many types of dendrimers have now been reported according to the type of intended application. 

An important consideration in dendrimer design is the ease of their preparation. Two main methods for dendrimer synthesis using divergent and convergent approaches were initially developed (Figure 8). In these approaches, the construction proceeds from the core to the shell or from the periphery to the core, respectively. These methods have since been elaborated upon and complemented by other synthetic methodologies such as orthogonal coupling, self-coupling assembly and solid-phase synthesis to meet the demand for new nanostructures and more efficient procedures that can be carried out cleanly, with good yields and on a useful scale [59].

Although their preparation is usually more demanding than that of normal polymers, dendrimers have the advantage that they are monodisperse compounds with a well-defined molecular weight. Their pharmacodynamic properties therefore display the reproducibility which is often preferred for biomedical applications. As a result of their special characteristics, dendrimers were investigated very early on for their potential use as imaging agents, tools in biological assays, drugs and as a means for drug delivery [59,61,62]. 

The special structural features of dendrimers, and the range of modifications that can be effected, enables them to act as targeted DDS with reduced drug doses and consequently reduced side effects. Dendrimers offer the possibility of encapsulating hydrophobic, hydrophilic or amphiphilic chemicals in their internal cavities via non-covalent interactions and, by acting as solubilizing agents or drug stabilizers, and by providing protective storage spaces for transport into the bloodstream, they can offer improved drug bioavailability. Dendrimers can also host substances on their periphery, where targeting groups or active biological compounds may be directly conjugated, either covalently or ionically, to the dendrimer surface by subsequently cleavable bonds. This approach provides a route to improved drug solubility and targeting through the appropriate engineering of the DDS such that it is sensitive to stimuli such as changes in pH, redox potential or enzymatic reaction. 

In addition, peripheral groups can also play a role in assisting transport across the BBB, and abundant experimental bibliographic data suggest that dendrimers are suitable molecule transporters for the efficient targeting and delivery of drugs to the brain. A rich array of innovative dendrimers have been designed through the specific functionalization of the dendrimer periphery (e.g., by -OH, -SH, -NH2, -COOH, azido and allyl groups) and the conjugation of carrier species such as carbohydrates, peptides, specific ligands or functional groups. Modification of the periphery also provides a means of exploiting the different active pathways for crossing the BBB by means of the endogenous molecules mentioned above. Functionalization with a nutrient such as glucose, maltose, or amino acids such as histidine has been utilized to trigger the carrier-mediated-transport mechanism to cross the BBB. Alternatively, functionalization with a positively charged group can be used to activate non-selective absorptive-mediated transcytosis, or the selective receptor-mediated-transport mechanism may be invoked by the anchoring of a ligand such as transferrin (Tf), sialic acid (SA) or lactoferrin (Lf) on the dendrimer [63,64,65].

These features are conveniently illustrated by the widely used polyamidoamine (PAMAM) system which is characteristic of amino-terminated dendrimers (Figure 9). PAMAM dendrimers, as well as the similar poly(propylene imine) (PPI) and poly-L-lysine (PLL) systems, have been widely accepted by biologists and medicinal chemists and have mainly been applied as solubility enhancers for hydrophobic molecules, the internal part of the dendrimer consisting of hydrophobic cavities which can entrap guest molecules at physiological pH, where both host and guest molecules are ionized. The larger the size or generation of the dendrimer, the greater the number of cavities, thus providing potentially better drug loading capacity. PAMAM dendrimers are relatively easily prepared in useful quantities, and the peptide skeleton in their branching amide bonds allows them to be manipulated according to standard protein chemistry protocols. They therefore represent one of the earliest commercially available classes of dendrimers to be applied as DDS [66]. 

One of the main drawbacks of unmodified PAMAM dendrimers is their cytotoxicity due to the protonation of primary amine groups on their surface at physiological pH. This leads to the disruption of cell membranes due to the cationic dendrimers stabilizing very strong electrostatic interactions with the negatively charged surface of the cell membrane, leading to its lysis. This toxicity is amplified for higher-generation dendrimers. It has therefore often been found to be beneficial to use lower generations in order to increase in vivo biocompatibility. Less toxic PAMAM dendrimers can also be obtained by functionalization of the peripheral amine groups by means of carboxylation, acetylation or PEGylation, resulting in charge neutralization or the creation of polyanionic half-generation PAMAM dendrimers with carboxylate groups on the surface (Figure 10). In spite of their potential toxicity drawbacks, however, it is worth noting that cationic commercial dendrimers have found potential application as nanocarriers for gene delivery, where their positively charged surfaces interact with the negatively charged nucleic acid [63]. 

## 5. Dendrimers in Alzheimer’s Disease: From Hypotheses to Therapeutic Strategies

Dendrimers offer great potential for non-invasive treatments of brain disorders [68,69]. Appropriate modifications enable them to penetrate the BBB, and they possess the advantages of other multifunctional molecules specifically designed to specific target cerebral dysfunctional mitochondria or activated microglia implicated in AD pathology. Numerous in vitro and in vivo studies also indicate that certain dendrimers interact with protein fragments involved in neurodegenerative diseases and that dendrimers can therefore sometimes play a dual role as DDS and as active agents per se. As a result, various therapeutic strategies against AD have been proposed which make use of dendrimers, and these will be described below in relation to the different hypotheses for the causes of AD.

### 5.1. PAMAM as DDS in AD Symptomatic Treatments

AD is a complex disease and, although potential monotherapies or combinations of cholinesterase inhibitor and uncompetitive antagonists of the N-methyl-D-aspartate (NMDA) receptor [5,6] have been investigated, none of them have so far been clinically very successful. Nevertheless, these drugs are very important since, by treating the symptoms, they contribute to an improvement of cognitive abilities. Drugs such as donepezil, galantamine and rivastigmine are FDA-approved formulations of therapeutic significance in AD (Figure 11). As cholinesterase inhibitors, they are thought to overcome the presynaptic cholinergic deficit, which causes memory and cognitive impairment and is related to the hypersecretion of acetylcholine and reduced choline absorption [70]. On the other hand, hyperexcitation of NMDA receptors induced by glutamate excess is responsible for Ca^2+^ overload and a cascade of events associated with apoptosis [71]. This has led to the use of uncompetitive NMDA receptor antagonist or glutamatergic drugs to reduce excitotoxicity. Memantine (MEM) is such an antagonist and forms part of an FDA-approved formulation for the symptomatic treatment of the disease. 

Tacrine (TAC) is another AChE inhibitor which, in the past, has been approved for the treatment of AD. Hepatotoxicity concerns, however, have led to the approval of this drug being withdrawn. In order to ameliorate the toxicity aspect, the use of generation 4 and 4.5 PAMAM dendrimers as tacrine DDS has been investigated [72]. In the TAC: G4.0 PAMAM system, the dendrimer was used unmodified, namely with amino terminal groups, whereas in the TAC: G4.5 PAMAM system, the dendrimer had a periphery of negatively charged carboxylate terminal groups. Although G4.0 PAMAM demonstrates known toxicity, it was chosen because by itself it interferes with the formation of β-sheet amyloid fibrils and with AChE activity. No cytotoxic or hemolytic effects were observed in vitro or ex vivo and both dendrimer/drug mixtures were found to be as effective as TAC itself, but with diminished hepatotoxicity and greater biocompatibility. 

The same PAMAM dendrimers have also been proposed for the delivery of carbamazepine (CBZ), a potent enhancer of natural autophagy processes involved in the disruption of protein aggregates [73]. The FDA has approved CBZ for the treatment of epilepsy, neuralgia and bipolar disorder but, when it is administered as a solid, it is often accompanied by multiple side effects, and it was anticipated that these could be minimized by the use of a dendrimer for the delivery of the drug. CBZ was encapsulated in the G4.0 and G4.5 PAMAM dendrimers, and both of the CBZ–dendrimer complexes tripled drug solubility without exhibiting hemolytic effects. Only the CBZ: G4.5 PAMAM system, however, displayed an in vitro reduction in cytotoxicity compared to that of free CBZ, thus highlighting the benefits of using neutral or negatively charged dendrimers compared to cationic ones.

Bio-functionalized dendrimers have been used as platforms for targeted drug delivery. For example, transferrin receptors such as lactoferrin (Lf) mediate iron transport across the BBB, and G3.0 PAMAM dendrimers, functionalized with Lf, have been investigated for targeted delivery of rivastigmine (RIV) or memantine (MEM) to the brain [65,74,75]. Lower-generation dendrimers were preferred for reasons of toxicity and biocompatibility. Studies have demonstrated the feasibility of combining brain targeting and delivery of a higher drug payload; a highly efficient drug delivery in both non-induced and AD-induced animal models was obtained and was correlated with improvements in cognitive responses. Additionally, Lf labeling of the surface of the positively charged G3.0 PAMAM dendrimer reduced its hemotoxicity, thus resolving one of the most difficult aspects of the development of nanocarriers for parenteral administration. Labeling with Lf also positively influenced the MEM release profile, and a correlation was made between the improvement of the overall bioavailability of the drug and the biodistribution of MEM in the brain of animal models. As a result, Lf conjugation has also been proposed for treatments of other severe brain diseases such as Parkinson’s disease, cerebral palsy or brain malignancies, as well as in drug co-delivery systems. 

### 5.2. Dendrimers as Fibril Disrupting Agents or Drugs per se 

Therapeutic alternatives to fibril-disrupting agents include either blocking the conversion of monomeric Aβ peptide into toxic oligomers or locking the β-sheet Aβ aggregates into nontoxic structures. 

PAMAM dendrimers have been demonstrated to inhibit the formation of amyloid deposits in incubated cells while at the same time exerting hydrolytic activity on pre-existing forms of toxic aggregates [53,76,77]. In systems modeling Aβ fibrillation, other cationic dendrimers such as G4-polypropyleneimine (PPI) (Figure 12) and G4-phosphorus dendrimers (CPD), or neutrally charged dendrimers such as morpholine-G3-acid-triethylene glycol dendrimers, have also been investigated [53,78]. All of these dendrimers affect the Aβ polymerization process by either accelerating or inhibiting fibril production. Dendrimers limit the lifetimes or the formation of the toxic species implicated in this process rather than the fibrils themselves. The effect observed, i.e., acceleration or inhibition, depends on the dendrimer to peptide ratio. Acceleration of fibril formation is observed at low ratios, while the rate of polymerization and the fibril content are reduced at higher ratios. The degree of inhibition of fibrillation is also proportional to dendrimer generation.

The process by which dendrimers affect amyloid formation depends on their intrinsic structure as well as on pH, which directly influences the charges of the interacting species. The effects of PAMAM and CPD on Aβ28, one of the model peptides for β-amyloid, were examined in the presence of heparin, a model for a negatively charged cell surface and also a fibril initiator [79]. It appears that the dendrimers are involved in a double electrostatic interaction with heparin and peptide. PPI dendrimers show a higher inhibitory effect on fibril formation at pH6 compared to pH5, which was explained by a decrease in the protonation of histidine residues at higher pH.

All of the studies outlined above concern in vitro model systems, and largely disregard the major drawback of amine-terminated PAMAM or PPI dendrimers, namely their toxicity in many of the cell lines used in amyloid toxicity assays. The design and development of glycodendrimers as potentially biocompatible systems has thus attracted much interest [80,81,82,83,84,85]. By modifying the surface groups of cationic dendrimers with oligosaccharides, strong electrostatic interactions are replaced by weaker hydrogen bonds.

G4 PPI dendrimers surface-modified with maltose (Figure 13) show a similar ability to interfere with the Aβ40 fibril formation process as PAMAM or CPD. At high dendrimer/peptide ratios, amorphous granular non-fibrillar aggregates are formed (GNA in Figure 14), and are found to be toxic to the neuronal cell lines examined. When the dendrimer/peptide ratio is low, clumped but non-toxic fibrils are produced [81]. 

The same dendrimers, and other related partially or non-maltose surface-modified dendrimers, have been used to provide dendrimers with neutral or positively charged peripheries for further in vitro and in vivo studies. Neutral or cationic G4 PPI dendrimers were demonstrated to be able to penetrate the cytoplasm in cultured cells. They also reached the brain in APP/PS1 mice, either by crossing the BBB or after intranasal administration, and a modification of the total β-amyloid burden was observed [81]. However, following chronic administration, memory impairment was not reversed, and application of the cationic dendrimer caused cognitive decline in non-transgenic mice. In addition, a recent study has shown that multiple in vivo administrations of maltotriose-modified PPI dendrimers induced adverse effects in rats, with the partially modified dendrimer proving to be more neurotoxic than the fully modified one. Cationic PPI glycodendrimers must therefore be modified before they can be used for medical purposes, and further studies on their possible neurotoxicological effects are likely to be needed [86]. It was found that, by modifying such dendrimers with histidine to form G4HisMal PPI dendrimers, their selective capacity to cross the BBB could be improved, as well as their bioavailability in vivo. The intranasal quarterly administration of the dendrimers also protected APP/PS1 mice from synapse and memory deficits, and it seems that the copper chelating ability of histidine could provide additional neuroprotective activity against ROS formation. The dendrimer seems to shield the synapses against soluble toxic Aβ oligomers. G4HisMal PPI dendrimer therefore appears as a potential and safe neuroprotective agent in transgenic mice [87].

It is appropriate to mention here some other interesting approaches using modified bio-conjugated PAMAM in the context of therapeutic efforts against AD. Small dendrimers with terminal units derived from trehalose or gallic acid (Figure 15) have been studied for their potential as inhibitors of protein aggregation [88,89]. Trehalose, a disaccharide, and gallic acid, a natural polyphenol, both interfere with Aβ nucleation and aggregation processes. These effects are enhanced by their multivalent presentation on dendrimers. The trehalose–PAMAM system contains multiple polyhydroxyl terminal units which, via strong hydrogen bonds, strongly inhibit amyloid cluster formation, whereas, in the gallic acid–PAMAM dendrimer, the gallate terminal residues induce the formation of non-cytotoxic aggregates and the disintegration of matured protein fibrils via both hydrophobic and H-bonding interactions. 

The β-amyloid peptides can also bind to cells via interaction through sialic acid (SA) residues found on the surface glycolipids or glycoproteins, and the removal of SA from the cell surface attenuates Aβ toxicity. SA–PAMAM dendrimers were prepared either by attachment via the carboxylic group or via the anomeric hydroxyl group, the latter being closer to the physiological mode of attachment. Τhe dendrimers were able to attenuate Aβ toxicity at approximately three orders of magnitude lower concentrations than the soluble sialic acid [64,90]. In another less recent study, the KLVFF peptide, which has been identified as the nucleation site for Aβ aggregation and corresponds to an amino acid fragment located in the central region of Aβ, was conjugated with a first-generation PAMAM dendrimer. The resulting molecule inhibits the aggregation of Aβ42 into fibrils in a concentration-dependent manner. More effective than KLVFF itself, the dendrimer promotes the disassembly of already formed aggregates. In a preliminary study which, however, does not appear to have been followed up, PAMAM dendrimers grafted with curcumin were also reported to inhibit amyloid aggregation and dissolve amyloid fibrils [91].

### 5.3. Dendrimers for the Targeting of Oligomers as AD Biomarkers 

As mentioned previously, the onset and progression of Alzheimer’s disease are thought to depend on toxic oligomer species. These have been widely described in vitro and characterized from homogenates of diseased brains, but their identification in vivo remains a challenge to which powerful imaging techniques (MRI, PET) can only provide an answer by detecting the presence or absence of non-toxic fibrillar structures. However, infrared microscopy (μ-FTIR) used over the last ten years for the in situ study of amyloid deposits and their structural characterization has made it possible to confirm the presence of non-fibrillar structures in the APP/PS1 transgenic mouse. Using this technique, the daily treatment of mice with G4HisMal-PPI dendrimers has been demonstrated to cause a reduction in fibrillar aggregation and the formation of non-fibrillar aggregates. These aggregates are found to be abundant in untreated APP/PS1 transgenic mice at an early stage of disease progression, in contrast to mice treated with the G4HisMal-PPI dendrimer in which the level of aggregation in general, i.e., fibrillar and non-fibrillar, is severely reduced. Globular dendrimers seem to interact directly through hydrogen or ionic bonds with the amyloid peptide or its aggregates, and indirectly through the chelation of copper ions by the histidine residues. They slow down the nucleation-dependent polymerization process and thus reduce the number of fibrils produced. 

The identification, localization and characterization of these non-fibrillar aggregates in the cerebral cortex of transgenic mice at early stages of AD progression underline the relevance of the pharmacological targeting of such species as AD biomarkers, and the use of non-invasive diagnostic techniques involving dendrimers could facilitate the design of more effective drugs [92]. 

### 5.4. Cationic Phosphorus Dendrimers and the Tau Hypothesis

The inhibition of tau abnormality has been considered to be one of the most promising therapeutic approaches to AD, since pathological aggregation has been correlated with the hyperphosphorylation of microtubule-associated protein (MAP)-Tau in neurons and microglia. It also seems that soluble oligomers of hyperphosphorylated tau, rather than fibrillar aggregates, are the most toxic assemblies of tau in AD, although the nontoxicity of amorphous aggregates remains uncertain. Cationic phosphorus dendrimers (CPD) (Figure 16) have been shown to exhibit both anti-amyloidogenic and anti-tau properties, inducing the aggregation of tau into amorphous rather than filamentous structures. The most striking effect is the ability of CPD to diminish the toxicity of prefibrillar Aβ28 species [78,93].

The synergistic action of CPD with conventional AD treatment has been investigated because the mechanisms involving Aβ toxicity in AD are related to Aβ’s disruption of cholinergic neurotransmission, as well as oxidative stress related to Aβ’s activation of microglia and disruption of the mitochondrial respiratory chain. As the surface of CPD dendrimers is not fully ionized at physiological pH, these dendrimers are less toxic than conventional amino-terminated dendrimers. However, their use is only possible in a limited range of dendrimer concentrations. CPD dendrimers have shown an ability to modulate the immune response and exhibit weak antioxidant properties without interfering with the action of AChE inhibitors. The authors point out that the transition from CPD to clinical evaluation will, however, require modification of the dendrimer surface to reduce its toxicity and increase its ability to cross the BBB.

### 5.5. Dendrimers and Neuroinflammation 

It is now widely believed that the neuroinflammatory process begins in the earliest phase of AD and can have positive and negative consequences. The brain immune system recognizes the abnormal accumulation of proteins as injurious stimuli, so it initiates a physiological reaction to defend the brain, which includes the activation of glial cells and the release of proinflammatory molecules [95]. 

Aβ oligomers and insoluble Aβ fibrils bind to the cell surface of microglia, making them sensitive to secondary inflammatory mediators and giving rise to an amplified inflammatory reaction. Microglia undertake the clearance of Aβ oligomers and fibrils by phagocytosis and also by surrounding plaques and fibrils to create a kind of physical barrier which prevents their spread. Two types of microglia cells with opposite activation phenotypes are present in the brain, generating either cytotoxic or neuroprotective effects. In the M1 or “proinflammatory” phenotype (a kind of stand-by mode), the release of neurotrophic factors decreases and inflammation and cytotoxicity are exacerbated. There is an impairment in microglia migration and, in cases of AD with notable chronic inflammation, it has been noted that microglia lose their ability to remove misfolded proteins effectively. On the other hand, in the M2 or “anti-inflammatory” phenotype, anti-inflammatory cytokines are released together with neurotrophic factors, leading to neuroprotective effects (e.g., downregulation, protection, or repair processes in response to inflammation). Thus, the early activation of microglia can give rise to an advantageous outcome, but when there is a chronic activation by Aβ, activation becomes detrimental and induces protracted inflammation and disproportionate Aβ deposition and hastens neurodegeneration. Targeting inflammation in general, and pathological gliosis more specifically, thus represents a promising and innovative therapeutic approach for several brain diseases. 

Dendrimers such as amino- or hydroxyl-terminated PAMAM have been demonstrated to present anti-inflammatory properties, and offer a potential means for the treatment of chronic inflammation when conjugated to different anti-inflammatory compounds [96,97]. PAMAM dendrimers with bile acid termini such as that shown in Figure 17 are examples of this type.

Although there is much evidence that indicates that the BBB is impaired in neurodegenerative diseases such as AD, it has been demonstrated that G4 hydroxyl-terminated PAMAM dendrimers (G4-PAMAM-OH) are able to cross the impaired BBB after systemic administration in numerous neuroinflammation disease models. This allows for the use of a non-invasive delivery route instead of highly invasive local delivery procedures, which have been used with other nanoparticles (e.g., poly-ε-caprolactone, PEG, and negatively charged PAMAM dendrimers) [9,98]. In fact, it has been demonstrated that G4-PAMAM-OH dendrimers accumulate in activated microglia without the need for any targeting ligands [99]. G4-PAMAM-OH are ideally suited to clinical translation due to their scalability and favorable in vivo safety profile.

Minocycline (Figure 18) is a biologically active compound, which has shown promise for the treatment of neurological diseases due to its combined anti-inflammatory and anti-oxidant properties. It has also demonstrated an ability to cross the BBB in a rabbit model with cerebral palsy. Mino was conjugated to the higher-generation dendrimer, G6-PAMAM-OH, which has the advantage of a longer blood circulation and has also been demonstrated to cross the impaired BBB. The Mino–dendrimer conjugate, D-Mino, enhanced the intracellular availability of the drug due to an excellent uptake into activated microglia, leading to the suppression of inflammatory cytokines and a reduction in oxidative stress. D-Mino facilitated minocycline’s crossing of the impaired BBB and exhibited targeting of the activated microglia at the site of injury [100].

N-acetyl-cysteine (NAC), a molecule demonstrating antioxidant and anti-inflammatory abilities, is not only able to directly scavenge for ROS but, as a glutathione precursor, can also stimulate the production of endogenous antioxidant glutathione, the latter preventing free radical formation, ameliorating mitochondrial dysfunction and decreasing oxidative stress. NAC has also been widely evaluated for its potential for the prevention and treatment of cognitive aging and dementia. However, it demonstrates poor bioavailability due to serum protein interactions, necessitating multiple high-dose administrations to observe neuroprotective effects. Additionally, NAC cellular internalization occurs through the cysteine-glutamate antiporter, which results in glutamate release and subsequent excitotoxicity and neuronal damage. Therefore, in an attempt to minimize these drawbacks, NAC has also been conjugated to G4-PAMAM-OH [101,102]. The resulting dendrimer, D-NAC (Figure 19), has been studied in different brain pathologies. Its systematic administration led to a dramatic improvement in motor function and neuroinflammation attenuation in a rabbit model for cerebral palsy, and was demonstrated to enable the normalization of reactive microglia in several animal models [103,104,105]. 

Another strategy that has been proposed is to target pathological gliosis. The pro-inflammatory M1 microglia play a critical role in the pathology of neurodegenerative diseases and the possibility to administer a drug via the BBB, which induces a change in the “M1 to M2” microglia phenotype, opening up new perspectives for the treatment of AD. Epidemiological studies have demonstrated that type 2 diabetes patients taking peroxisome proliferator-activated receptor-γ (PPARγ) agonists such as rosiglitazone and pioglitazone are at a reduced risk for developing AD [106]. Although these FDA-approved agonists induce an interesting M1 to M2 phenotype shift in microglia via a reduction in ROS secretion, they proved to be unsuccessful in phase III clinical trials for AD due to their poor drug transport across the BBB. A similar compound, tesaglitazar (Tesa, Figure 18), which had also been developed for the treatment of type 2 diabetes, has also attracted interest in connection with AD. Although it presents a satisfactory safety profile, Tesa failed in phase III clinical trials for type 2 diabetes due to its dose-dependent toxicity, and so the conjugation of Tesa to a dendrimer via its labile carboxylic function was considered in order to resolve issues such as controlled release, a decrease in drug dose and targeted delivery to activated microglia [107]. The design and synthesis of D-Tesa (Figure 20), a G4-PAMAM-OH drug conjugate, was undertaken with two strictly connected objectives, namely (a) the delivery of the drug, after systemic administration, to induce a microglia M1 to M2 phenotype shift and (b) a reduction in the secretion of neurotoxic substances, leading to an anti-inflammatory state and a subsequent increase in the degradation and phagocytosis of pathogenic proteins in the brain.

D-Tesa also offers a great improvement in water solubility, thus avoiding the use of toxic excipients in the drug formulation. The dendrimer’s ability to freely cross the BBB to access microglia in vivo enabled a reduction in the dose required to achieve a therapeutic effect, thus further ameliorating toxicity issues. When Tesa is conjugated to the dendrimer via an intracellularly cleavable labile ester bond, about two-thirds of the drug is released within the first 48 h under lysosomal conditions. D-Tesa was shown to possess the ability to induce a M1 to M2 phenotype shift in microglia in vitro, enhancing phagocytosis of fluorescently labeled β-amyloid. Induction of the phenotypic change by D-Tesa exceeded that of Tesa itself and was also accompanied by a reduction in the secretion of nitric oxide and an increase in the expression of α-degrading enzymes of β-amyloids and also of β-amyloid phagocytosis. Thus, D-Tesa, when administered at an appropriate stage of disease progression, provides a potential agent to fight many neurological conditions and also represents an interesting tool for a better understanding of the role of the microglia phenotype in neurodegenerative diseases such as AD.

Peptide therapeutics are agents that show significant efficacy in inhibiting protein–protein interactions. However, peptides by themselves have a number of drawbacks associated with their size and high polarity, which can lead to poor pharmacokinetics and bioavailability. Dendrimer–peptide conjugates (DPC) have therefore received particular attention. A reactive oxygen species-sensitive dendrimer–peptide conjugate was designed to target the microenvironment typically found in AD and inhibit inflammatory responses at an early stage of the disease. By modifying the targeting peptide, a nanoplatform was constructed which had the ability to efficiently deliver peptides to Aβ-infected neuronal regions while promoting the restoration of the antioxidant capacity of neurons via the activation of the nuclear factor signaling pathway (Nrf2). This multi-targeting strategy approach thus combined the synergistic function of scavenging ROS, with the aim of promoting phagocytosis of Aβ locally, and the normalization of the degree of activation of glial cells such as microglia. An improvement in cognitive function in APPswe/PSEN1dE9 mouse models was observed, and this approach appears to be of great interest in the early treatment of AD [108]. 

Compounds able to control glial activation through a combination of neuroprotective and anti-inflammatory effects therefore constitute a pool of possible therapeutic drugs for further study. However, the effect of therapeutic interventions targeting glial cells depends on reaching the perfect balance between attenuating deleterious effects and maintaining the existing beneficial mechanisms of brain defense.

### 5.6. Dendrimers and Mitochondrial Dysfunction 

In light of all the hypotheses previously discussed in connection with the pathogenesis of AD, it makes sense to target mitochondrial dysfunction therapeutically. Indeed, mitochondria appear to at least mediate, if not trigger, pathological molecular cascades, and mitochondrial and amyloid cascades may coexist. As mentioned previously, a secondary mitochondrial cascade may mediate damage caused by a primary amyloid cascade, or amyloid produced as part of a primary mitochondrial cascade could itself cause harm. This could have important implications for future therapeutic developments [109].

Addressing a primary mitochondrial cascade would likely require unique strategies focusing on preventing age-related decline in mitochondrial function, for example through exercise or diet. Pharmacological approaches that enhance aerobic or other aspects of mitochondrial function, or overall cell bioenergetics, could also prove to be beneficial. Investigators who favor the secondary mitochondrial cascade hypothesis are currently developing interventions that would ideally interrupt very specific parts of an Aβ-dependent mitochondrial cascade. Their efforts include the deployment of small molecule inhibitors of Aβ–mitochondria interactions or interventions intended to block Aβ-induced changes in mitochondrial physiology [110,111]. Preventing an Aβ-induced increase in mitochondrial fission represents such an approach. In contrast with studies on the generation of ROS induced by mitochondrial dysfunction, studies on inflammatory responses associated with mitochondrial dysfunction are very scarce. Although mitochondrial components have been described as causal factors of inflammation (see Section 2.5), they are also suitable therapeutic targets for the regulation of neuroinflammation. Indeed, inhibiting mitochondrial inflammation or maintaining functional mitochondria through MQC can reverse many of the symptoms observed in the AD model. Thus, mitochondrial inflammation is a valuable diagnostic target and requires further study as an emerging therapeutic target for the treatment of AD [109]. 

The development of therapeutic strategies that target mitochondrial dysfunction through drug delivery to mitochondria and/or improve mitochondrial bioenergetics using antioxidant agents has been proposed. However, mitochondrial drug delivery is a difficult task due to the complex organelle membrane structure that surrounds the mitochondrial core. The membrane contains three parts: two key cellular barriers, the outer and the inner mitochondrial membrane (OMM and IMM) separated by the intermembrane space, and the core of the mitochondria or matrix [45]. Since the OMM does not display any membrane potential compared to the IMM that maintains a highly negative internal membrane potential, this first barrier is relatively weak. Molecules pass freely back and forth through the OMM via passive concentration-driven diffusion into the intermembrane space. In contrast, IMM has a more complex structure, and is highly and rigidly folded. It displays small transition pores, which separate the intermembrane space from the mitochondrial matrix, making it extremely difficult for small molecules to cross through. In addition, due to its high membrane potential, the IMM blocks the passage of polar and charged molecules into the mitochondrial matrix. Finally, the matrix contains an abundance of voltage-dependent anion channel (VDAC) transporting molecules less than 5000 mw into the transmembrane space between the OMM and IMM. It is noteworthy that gold nanoparticles of 3nm but not of 6nm can be transported through VDACs. 

The use of amphiphilic moieties to cross both membranes and to accumulate in the mitochondrial matrix is required, and mitochondrial targeting molecules such as delocalized lipophilic cations have been used. For example, the concentration of triphenylphosphonium (TPP) moieties, which exhibit a large hydrophobic surface, increases at the IMM five to tenfold compared to the extracellular environment, and a hundredfold in the mitochondria compared to the cytoplasm. Numerous therapeutic agents, including antioxidants, have thus exploited the properties of TPP to target the mitochondria [46,112]. 

Although efforts based on nanotechnology targeting mitochondria in diseased brain cells are still limited due to the lack of structural components to reach the mitochondrial matrix, several approaches aiming to restore mitochondrial function can be observed. A direct drug delivery strategy to mitochondria involves either exploiting the biophysical properties of the IMM or the integration of mitochondrial translocation ligands onto the surface of nanoparticles. 

PAMAM dendrimers with positively charged surface groups have been demonstrated to interact with cellular components (nucleus, mitochondria, enzymes, endosomes, cell membrane) with a resulting outstanding cellular internalization potential which makes them excellent carriers for cellular and organelle delivery [45,113]. TPP conjugated G4-PAMAM dendrimers demonstrated cellular internalization as well as significantly increased mitochondrial targeting at low TPP dendrimer surface densities (8%). With a doubling of TPP density, an increase in cellular internalization accompanied by pronounced cytotoxicity has been reported but, interestingly, the introduction of PEG linkers attenuated toxicity without affecting nanocarrier targeting ability [112]. However, neutral hydroxyl-terminated PAMAM dendrimers with a superior safety profile have been preferred as potential nanocarriers due to their ability to cross the impaired BBB, and they have been demonstrated to selectively accumulate at the site of inflammation in activated glia with minimal accumulation in healthy brain tissue [45,113].

As mentioned above, N-acetyl cysteine (NAC) has been considered as a neuroprotective agent and, when delivered in a targeted manner, it has also shown significant potential for clinical use as an antioxidant agent. Multifunctional hydroxyl-terminated PAMAM dendrimer conjugates have been developed for the in vivo targeting of mitochondrial function of activated microglia and macrophages in traumatic brain injury. The dendrimer surface is partially modified with TPP to confer intracellular mitochondrial targeting to complement the favorable microglia targeting potential and cell internalization properties of the unmodified dendrimers (Figure 21). Subsequent NAC conjugation to TPP-modified dendrimers (TPP-D-NAC) demonstrated enhanced efficacy against mitochondrial oxidative stress with significantly greater localization into mitochondria than for unmodified dendrimers [113]. 

In conclusion, the mitochondrial targeting capability of these dendrimers has been translated into enhanced efficacy in improvements of markers of oxidative stress, including mitochondrial superoxide production and transmembrane potential. In terms of oxidative stress-induced cytotoxicity, TPP-D-NAC performs significantly better in recovering cell viability compared to both D-NAC (see Section 5.5) and freely administered NAC, and has the potential to treat oxidative stress in the area of acute traumatic brain injury in vivo. Although this system was not examined in neurodegenerative diseases, these very interesting results suggest that TPP-D-NAC may significantly improve the efficacy of D-NAC in CNS disease pathologies with significant oxidative stress-induced cell death, such as traumatic brain injury [114], Alzheimer’s disease, and hypoxic ischemia.

Besides the major hypotheses that have already been covered, AD presents mitochondrial and bioenergetics alterations which could contribute to the development or even the progression of the disease [41]. Insufficient ATP production induced by proteasome malfunction or oxidative stress resulting from mitochondrial DNA mutation or the generation of ROS can lead to cell apoptosis and neurodegeneration [47,115]. Indeed, the brain is highly susceptible to oxidative imbalance of the equilibrium between the generation and detoxification of physiological ROS [44]. Subsequent oxidative stress-mediated damage to biomolecules has been extensively reported in AD and is thus generally assigned a critical role in the disease [116]. The progressive impairment of mitochondrial function implicated in aging seems to be the main source of ROS generation and a major target of oxidative damage [117], with the interaction between oxidative stress and mitochondrial dysfunction playing an important role in the pathogenesis of AD. There is also a question mark concerning whether mitochondrial dysfunction contributes to NFTs and β-amyloid plaque formation or the opposite. The anteriority of a mitochondrial cascade to an amyloid one, namely whether mitochondrial dysfunction precedes the appearance of Aβ or vice versa, is still under debate. Nevertheless, the available data seem to support the possibility of both primary and secondary mitochondrial cascades, and thus both currently represent AD therapeutic targets. Consequently, the targeting of mitochondrial dysfunction needs to be given serious consideration.

## 6. Conclusions

Although many therapeutical approaches to AD are based on the main hypotheses that have been proposed concerning its causes, it has also become apparent that it is important to investigate alternate avenues by which AD develops, since this has a direct impact on strategies for the development of potential treatments. Besides the major hypotheses that have been already covered, AD presents mitochondrial and bioenergetic alterations, as well as pathological neuroinflammation that could contribute to the development or even the progression of the disease from an early stage through the cross-talk between inflammatory response and brain immune cells’ response, and these are also aspects currently under investigation. Numerous hypotheses in addition to the proteinopathies such as the amyloid and tau hypotheses have been put forward, and the relative importance of various potential causes of AD has become increasingly clear. This, in turn, will influence the effectiveness of any proposed treatments, which optimally should be personalized according to factors such as age, genetics, risk factors, and timing of diagnosis and treatment. For example, atherosclerosis or cardiac dysfunction may appear in one AD patient but be absent in another. As Frenkel (2020) noted, there is the potential of an “arsenal of drugs that can be used to personalize medicine for each patient in the spectrum of diseases associated with AD” [118]. This approach, however, still faces many challenges, such as determining which “cocktail” of drugs to use and how drugs interact with one another, but nanotechnological constructs offer one possible route to personalize therapeutic protocols. Current therapeutic strategies focus on targeting Aβ but their clinical efficacy in advanced AD is compromised by the deleterious effects of plaques on the CNS. Thus, multi-targeting of an early-stage AD microenvironment has appeared to be of great interest.

A number of types of nanomaterials have been proposed and investigated for their potential in therapeutic approaches to the treatment of AD and each of them has certain advantages and disadvantages. It is beyond the scope of this review to provide a comparative assessment of these approaches, but some relevant information is, however, available in a number of recent reviews [10,52,54,98]. Dendrimers are not always the most economically attractive and readily produced type of nanomaterial but, with the variety of ways in which their properties and, in particular, their capacity to act as multi-targeting agents which combine synergistic functions can be fine-tuned, they show great promise and the question now to be answered is whether this promise can be realized. Improved methods of dendrimer production which are being developed will undoubtedly facilitate the more economical production of dendrimers, but another way forward to overcome the drawbacks of dendrimers and other types of nanomaterials could be through the design of hybrid materials which combine the advantages of the individual components. Liposomes, for example, are widely accepted and readily produced and generally have good loading capacities and low toxicity. Liposomes are also known to be able to pass through the BBB. It can be envisaged that their effectiveness and efficacy would benefit significantly by partial modification with dendrimers possessing targeting and imaging capabilities. It is expected that interest in systems such as these will become greater in the coming years due to the increased emphasis on personalized medicinal approaches. 

## Figures and Tables

**Figure 1 pharmaceutics-15-00898-f001:**
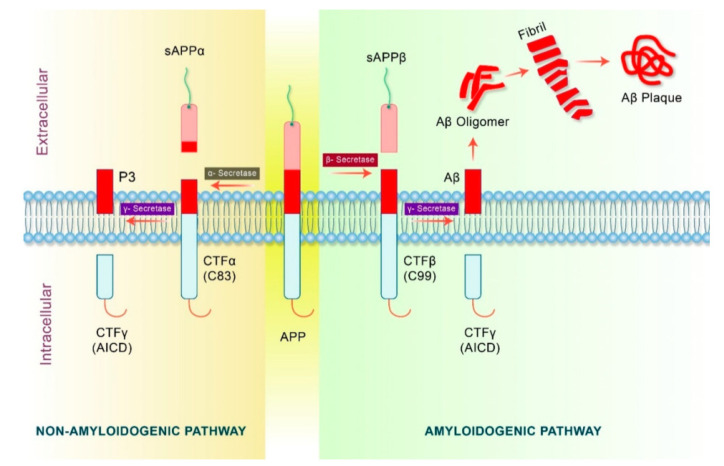
APP processing pathways in AD. Reprinted from [14].

**Figure 2 pharmaceutics-15-00898-f002:**
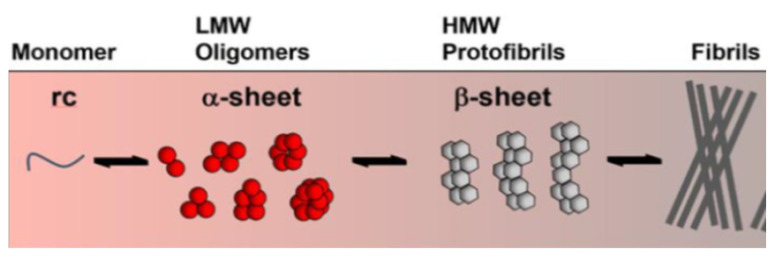
Aβ aggregation progress. Reprinted and adapted from [8].

**Figure 3 pharmaceutics-15-00898-f003:**
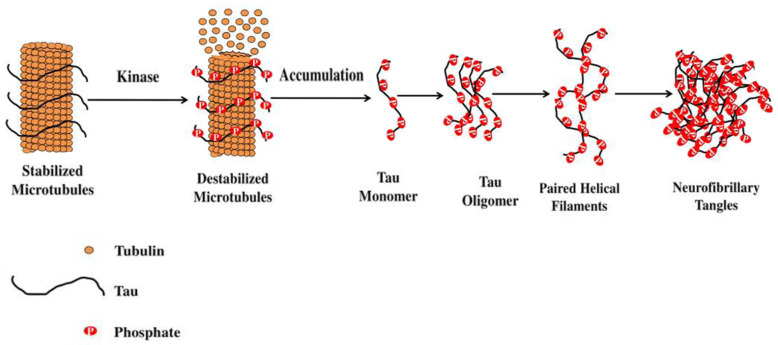
Stabilization of microtubule-associated tau protein is controlled by kinases. Reprinted from [29].

**Figure 4 pharmaceutics-15-00898-f004:**
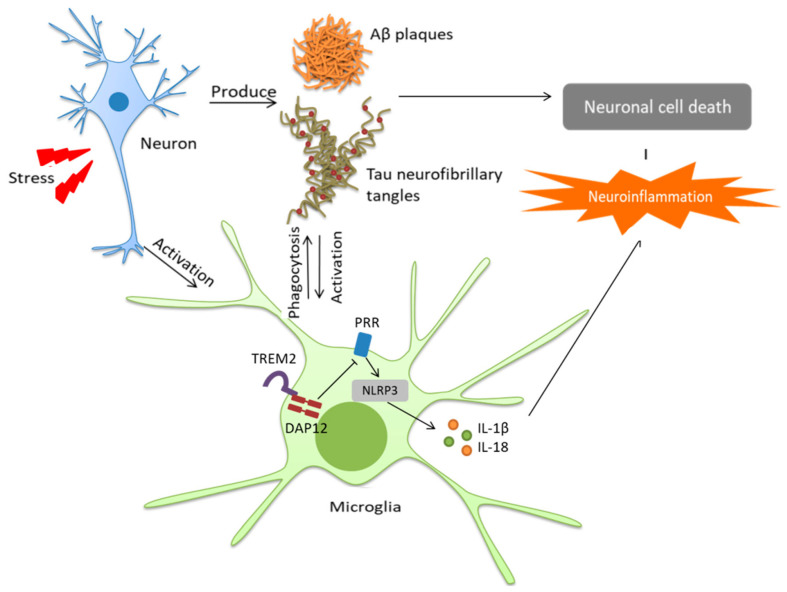
Microglia-induced neuroinflammation in Alzheimer’s disease. Reprinted from [34].

**Figure 5 pharmaceutics-15-00898-f005:**
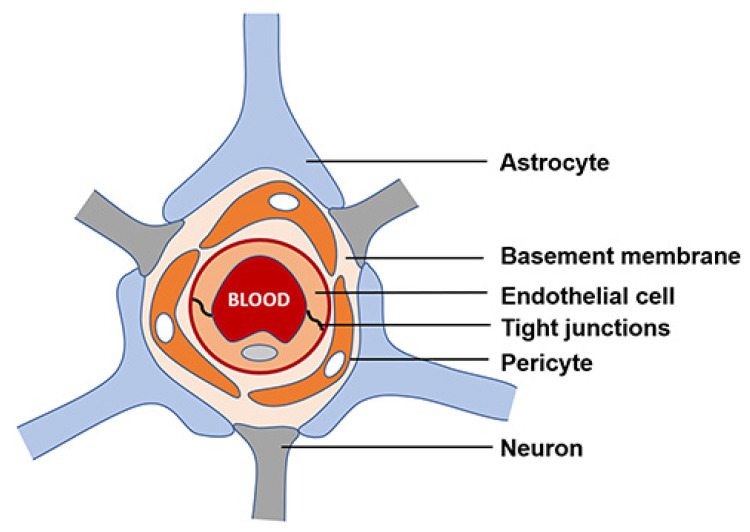
Schematic representation of the BBB. Reprinted from [51].

**Figure 6 pharmaceutics-15-00898-f006:**
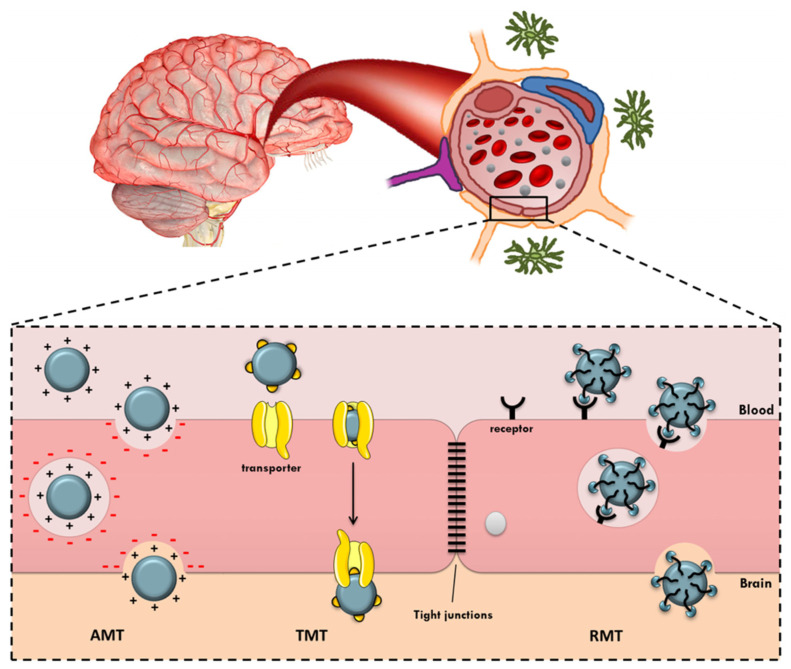
Mechanisms for transport across the BBB. Reprinted from [52].

**Figure 7 pharmaceutics-15-00898-f007:**
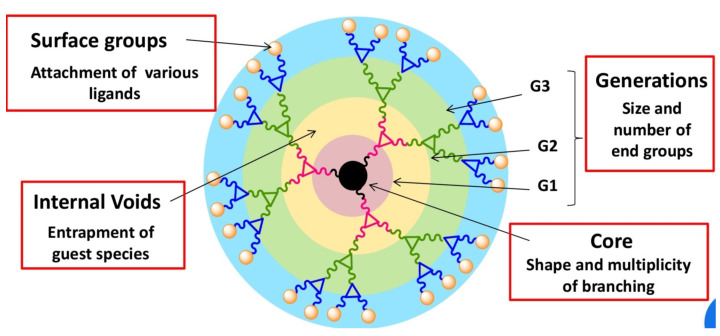
General features of dendrimer architecture. Reprinted from [58].

**Figure 8 pharmaceutics-15-00898-f008:**
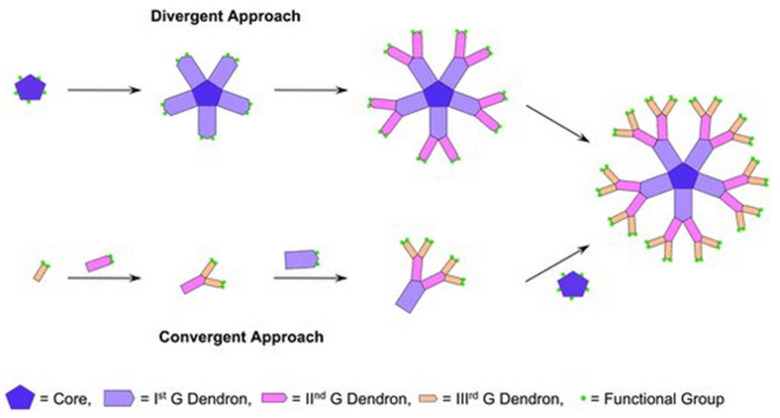
Divergent and convergent approaches for the synthesis of dendrimers. Reprinted from [60].

**Figure 9 pharmaceutics-15-00898-f009:**
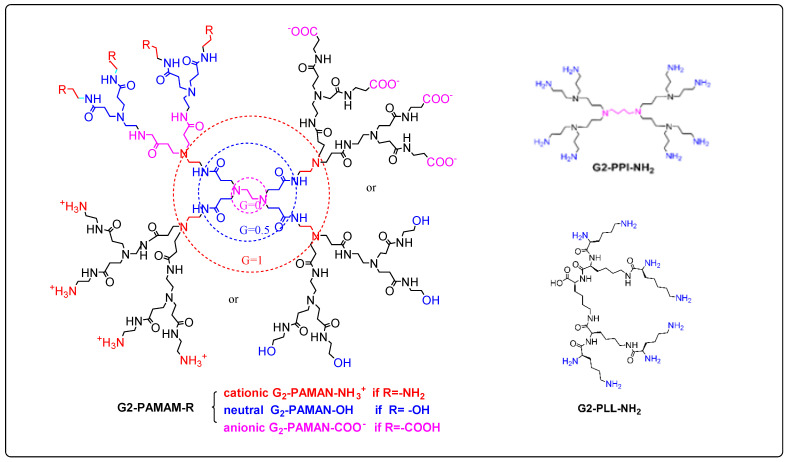
Schematic representation of generation 2-PPI, -PLL and NH_2,_ -OH or –COOH PAMAM dendrimers.

**Figure 10 pharmaceutics-15-00898-f010:**
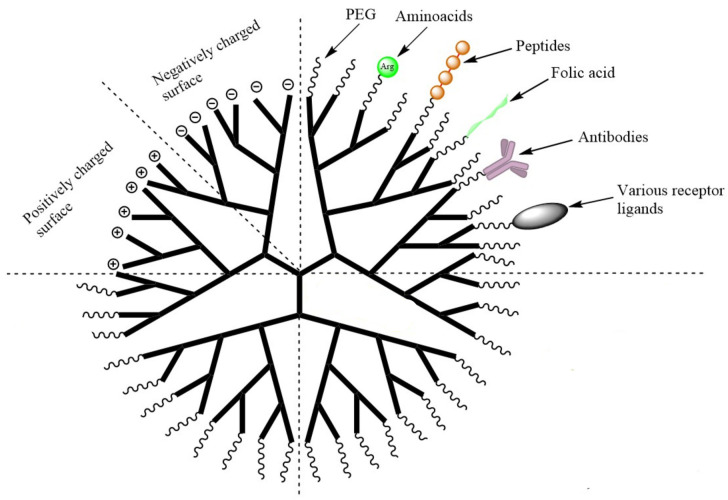
Functionalization of the dendrimer periphery. Reprinted and adapted from [67].

**Figure 11 pharmaceutics-15-00898-f011:**
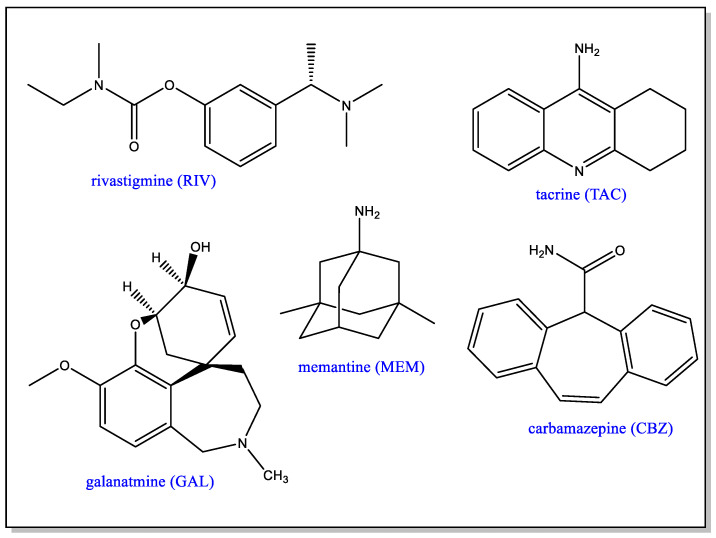
Drugs approved or under investigation for use in the treatment of AD.

**Figure 12 pharmaceutics-15-00898-f012:**
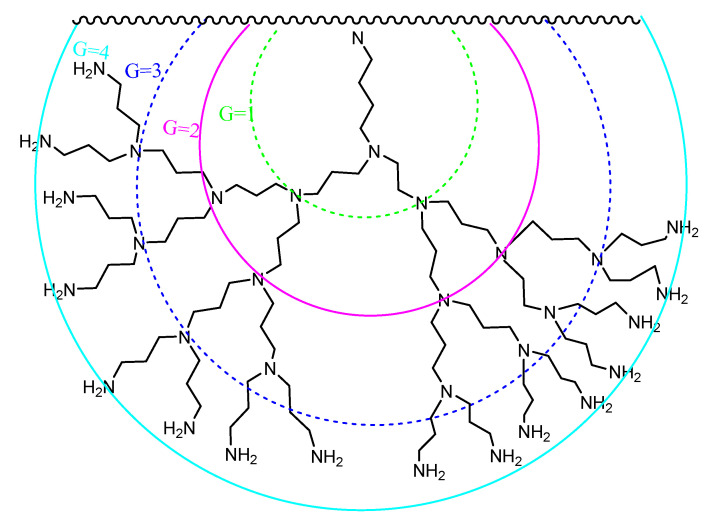
G4-PPI-NH_2_ dendrimer.

**Figure 13 pharmaceutics-15-00898-f013:**
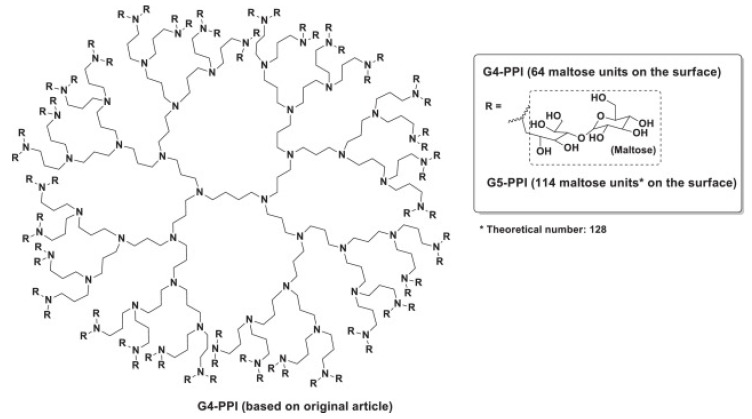
G4-PPI dendrimer with maltose substituted periphery. Reprinted with permission from [53].

**Figure 14 pharmaceutics-15-00898-f014:**
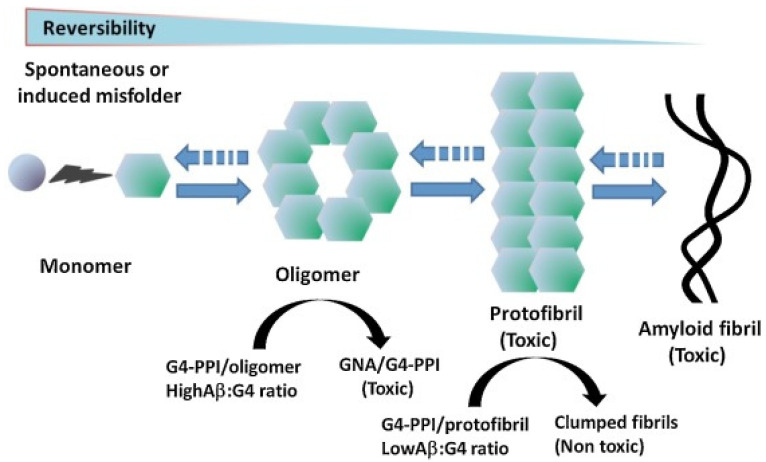
Schematic representation of the interactions between G4-PPI-maltose dendrimer and Aβ(1-40). GNA toxic granular non-fibrillary aggregates. Reprinted with permission from [53].

**Figure 15 pharmaceutics-15-00898-f015:**
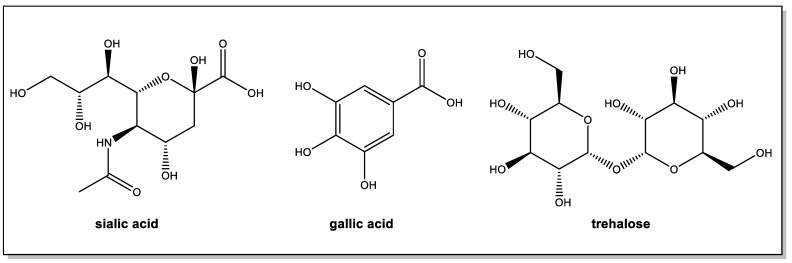
Polyhydroxy terminal groups in bio-conjugated PAMAM dendrimers.

**Figure 16 pharmaceutics-15-00898-f016:**
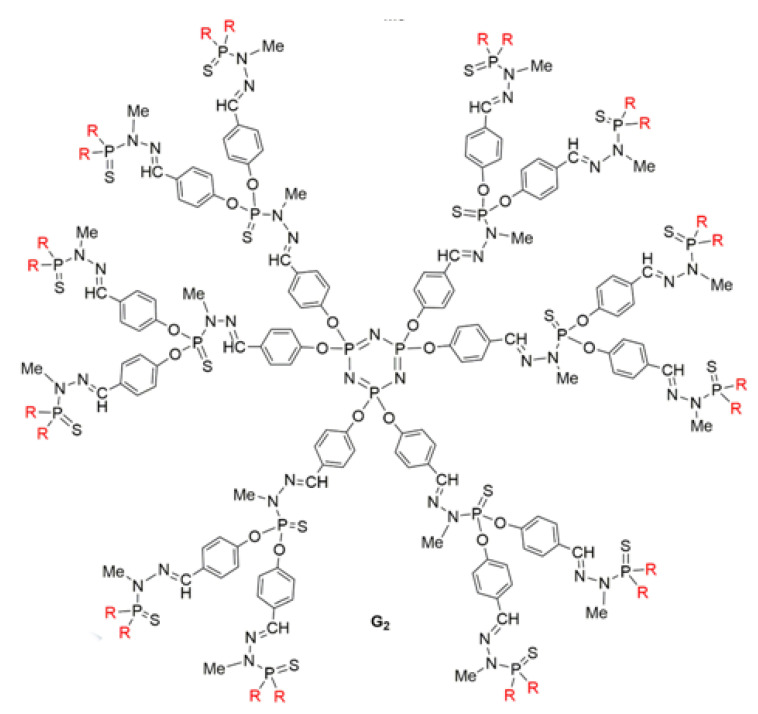
Cationic phosphorus dendrimer. Reprinted from [94].

**Figure 17 pharmaceutics-15-00898-f017:**
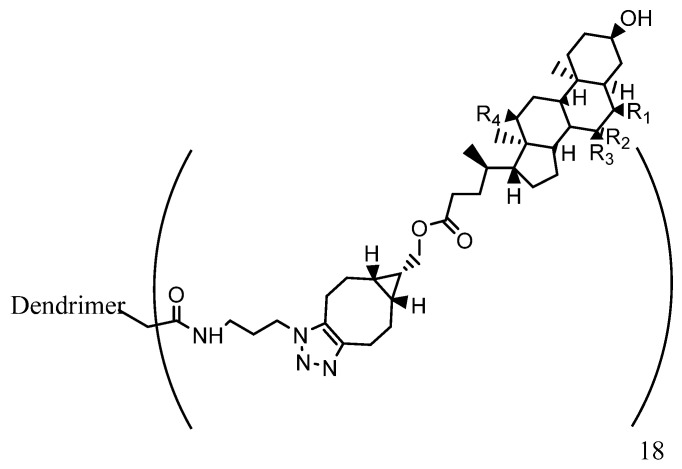
Bile acid-terminated dendrimer.

**Figure 18 pharmaceutics-15-00898-f018:**
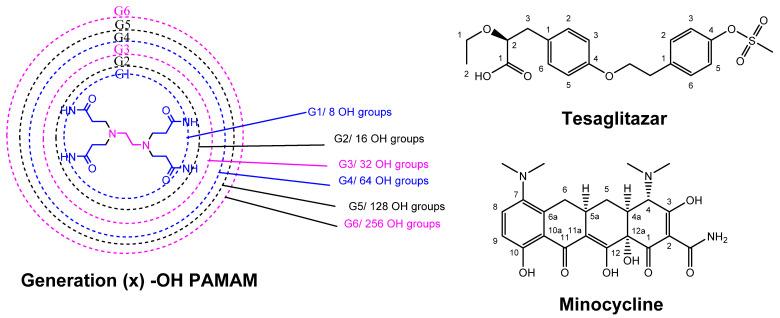
Structures of tesaglitazar, minocycline and hydroxyl–PAMAM dendrimers.

**Figure 19 pharmaceutics-15-00898-f019:**
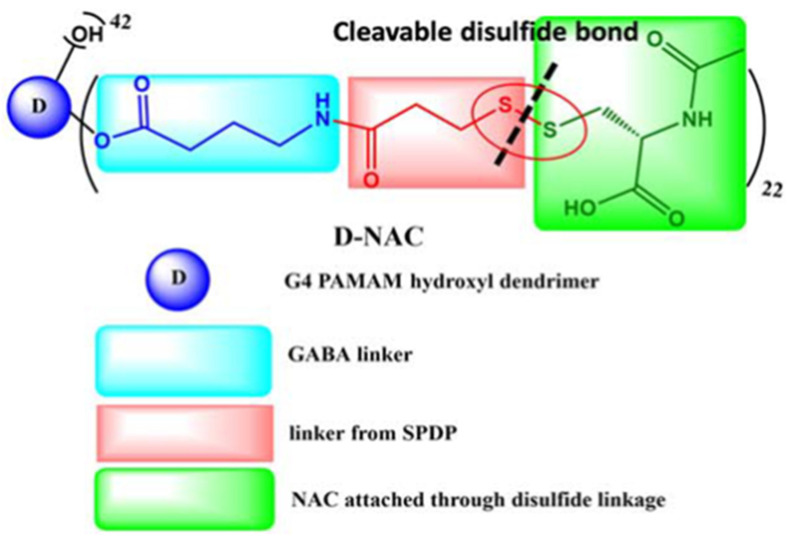
Structure of D-NAC dendrimer. Reprinted and modified from [102].

**Figure 20 pharmaceutics-15-00898-f020:**
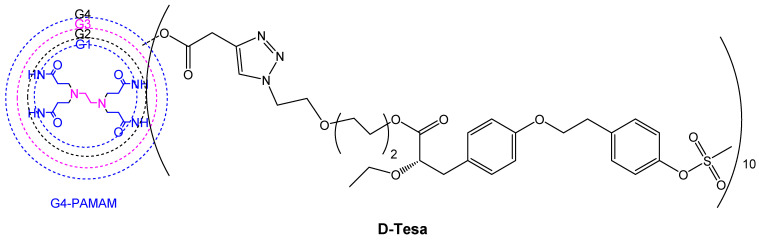
Dendrimer–tesaglitazar conjugate.

**Figure 21 pharmaceutics-15-00898-f021:**
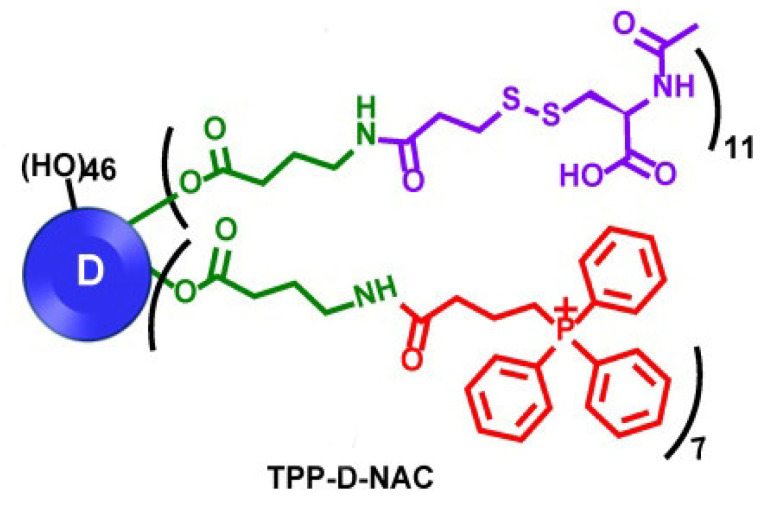
Triphenylphosphonium-N-acetylcysteine-modified hydroxyl–PAMAM dendrimer. Reprinted from [113].

## Data Availability

Not applicable.

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
