# Peer review of "Dendrimers in Alzheimer’s Disease: Recent Approaches in Multi-Targeting Strategies"

_pharmaceutics, 2023, doi:10.3390/pharmaceutics15030898_

Round 1

Reviewer 1 Report

This a nice review that first outlines what Alzheimer’s disease (AD) is, then the various hypotheses relating to the development of AD are commented and finally there is a description of example of dendrimer-based systems for therapeutic interventions of potential use in AD. The manuscript is timely. Besides, the work is very complete, and the manuscript is easy to read and clear. I would suggest acceptance after some minor correctios:

- The quality of the figures can be improved (at least in the manuscript I have read).

-  Figure 19. The dendrimer is not complete (some NAC groups are cut)

- The authors can work a bit more on the conclusions, for instance including some additional comments on the future research using dendrimers in AD and how dendrimers compare with other nanoparticles for AD treatment.

Author Response

  1. We have made modifications to figures that were considered to be on insufficiently high quality. The modified figures have been inserted into the manuscript and are also provided in the new compressed file uploaded via the website.
  2. The conclusion has been elaborated with a brief remark concerning the use of dendrimers in relation to other nanomaterials and also with a new paragraph on potential future developments that could be expected.

Reviewer 2 Report

Dear Authors

Congratulations for the nice effort, very beautifully explained the dendrimers-based drug delivery system for the AD. However, I am encouraging for the personalized approach which could be proved well before in-vitro system. Overall, the manuscript is well written and would be a valuable addition to this journal.

Please go through the whole paper and correct the duplication of the words and minor grammatical mistakes only.

Author Response

We would like to thank the reviewer for the very valuable suggestions and, as a result, we have made the revisions to the manuscript.

The text has been double-checked by a native English speaker and a number of minor corrections have been implemented.

Reviewer 3 Report

The review titled "Dendrimers in Alzheimer’s Disease: Recent Approaches in Multi-targeting Strategies” is well articulated, however few revisions are needed. 

1) The quality of Figures 3, 4 and 10 needs to be improved.

2) Line 45. The abbreviation “CNS” must be written in full and the acronym given in round brackets.

3) Line 140. To give a more complete picture of the action mechanisms underlying the toxicity of Aβ oligomers, the channel hypothesis should be briefly discussed (Kagan BL et al 2002). Numerous studies in the literature show the incorporation of Aβ oligomers into the membrane and the formation of the channel whose physiological characteristics have been studied (Arispe et al. 1993; Quist et al. 2005; Davis et al. 2010; Meleleo D et al. 2013, Bode DC et al. 2017; Meleleo D et al. 2019; Meleleo D et al. 2020. These works should be cited.

4) Line 205. In the sentence “The actual process by which HSV-1......", "in involved" must be replaced with "......is involved".

Author Response

  1. We have made modifications to figures that were considered to be on insufficiently high quality. The modified figures have been inserted into the manuscript and are also provided in the new compressed file uploaded via the website.
  2. Additional text has been added to Section 2.1 which outlines the ion channel variation of the amyloid hypothesis. Six new relevant references have been inserted.
  3. The CNS abbreviation has been clarified.
  4. The text has been double-checked by a native English speaker and a number of minor corrections have been implemented.